# Integrated Transcriptome and Untargeted Metabolomic Analyses Revealed the Role of Methyltransferase Lae1 in the Regulation of Phospholipid Metabolism in *Trichoderma atroviride*

**DOI:** 10.3390/jof9010120

**Published:** 2023-01-14

**Authors:** Yanxiang Shen, Yiwen Zhang, Hui Zhang, Xinhua Wang, Jie Chen, Yaqian Li

**Affiliations:** 1School of Agriculture and Biology, Shanghai Jiao Tong University, Shanghai 200240, China; 2State Key Laboratory of Microbial Metabolism, Shanghai Jiao Tong University, Shanghai 200240, China; 3Agriculture (South), Ministry of Agriculture, Shanghai 200240, China

**Keywords:** *Trichoderma atroviride*, methyltransferase *lae1*, RNA-sequence transcriptome, LC-MS/MS metabolome, phospholipid metabolism

## Abstract

The putative methyltransferase Lae1 is a global regulator in *Trichoderma,* which modulates the expression of secondary metabolite gene clusters, possibly via chromatin remodeling. Here we aimed to explore the specific transcription and metabolites profiles regulated by Lae1 in *T. atroviride* 23. Comparative transcriptomics and metabolome analyses between the *lae1* deletion (Mlae1) and over-expressing (Olae1) mutants were performed using RNA sequencing and QTOF-UPLC-MS techniques. In total, 1344 unique differentially expressed genes (DEGs) and 92 metabolites were identified across three strains. The significantly altered metabolic profiles revealed that the *lae1* gene modulates central carbon metabolism, amino acid metabolism, secondary metabolism, and phospholipid metabolism. The effects of *lae1* on phospholipid metabolism were further explored, and the findings showed that *lae1* modulates the composition and function of cell membranes and other metabolic activities, including the phosphotransferase system (PTS) and biosynthesis of secondary metabolites (SM). Phospholipid metabolism is related to energy metabolism, signal transduction, and environmental adaptability of microorganisms. These data showed that *Lae1* affects the primary metabolites, phospholipid, as well as the regulation of secondary metabolites in Trichoderma. This study could potentially provoke in-depth investigations of the Lae1-mediated target genes in phospholipid synthesis. The Lae1 may act as a novel target that is associated with disease defense and drug development in the future.

## 1. Introduction

*Trichoderma* species are filamentous fungi commonly found in soil, decaying wood, and plant rhizosphere. These species have been widely used as biocontrol agents owing to their defensive functions against plant pathogens [1]. *Trichoderma* fungi can effectively inhibit aerial and soil-borne diseases, especially root rot caused by *Fusarium* spp., *Rhizoctonia* spp., and *Pythium* spp. [2,3]. Their antimicrobial activity is associated with the secretion of cell-wall-degrading enzymes and the production of antifungal substances, including antibiotics, volatile metabolites, terpenes, peptaibols, and piperazine derivatives [4]. Moreover, *Trichoderma* fungi are effective plant growth promoters (PGPs) in several crops, including Solanaceae, cucurbits, and ornamentals [5].

In *Aspergillus nidulans*, a velvet complex affecting the production of several SMs and participating in light-responding development regulation comprises LaeA, VeA, and VelB proteins [6]. LaeA was initially identified as an *Aspergillus* nuclear protein that regulates morphogenetic development, secondary metabolism, and antibiotic production in various filamentous fungi [7]. Studies report that the laeA protein regulates the synthesis of secondary metabolites by catalyzing the transformation of heterochromatin to euchromatin [8,9]. Transcriptional profiles of the *laeA* gene in *Aspergillus nidulans*, *A.fumigatus,* and *A.niger* show that laeA regulates several genes in SM clusters. These genes, in turn, affect transmembrane transport, cell wall biogenesis, transcription regulation, and carbohydrate metabolism [10,11,12]. Some studies report that LaeA modulates developmental events, such as conidiation in numerous fungal strains [13,14,15,16].

Lae1 is the *Trichoderma* ortholog of *Aspergillus* laeA. It was initially reported as the expression product of polysaccharide hydrolytic enzymes [17]. *Lae1,* unlike *laeA,* does not affect the expression of polysaccharide hydrolase through histone methylation [17]. However, it affects the gene cluster of secondary metabolites in *A. nidulans* [18]. Moreover, *lae1* is not involved in the methylation of H3K4 or H3K9 in *T. reesei* [19]. Loss of *lae1* genes results in the downregulation of gene clusters responsible for synthesizing the secondary metabolite, bikaverin, in *Fusarium verticillioides* [20]. Some studies found that most genes affected by *lae1* are not clustered in the genome [21]. Recent studies report that the *lae1* gene plays an essential role in regulating GA (gibberellins) biosynthesis, peptaibols production, and tenuazonic acid production [22,23,24].

The entire profile of metabolites in an organism at a certain moment is referred to as the metabolome. Metabolomics is the analysis and study of the metabolome using different techniques. Metabolic profiling approaches are powerful tools used to study the effect of gene deletion on secondary metabolite formation and to elucidate the functions of the deleted gene clusters. Mass spectrometry (MS) coupled with different chromatographic techniques is an important approach in metabolomics owing to the high sensitivity, good selectivity, wide application range, and provision of important information. LC-MS is characterized by high sensitivity and high robustness. Metabolic characteristics of different samples can be compared using the LC-MS technique through comprehensive and unbiased high-coverage detection of all compounds in the samples.

The present study sought to compare transcriptomic and metabolomic variations between wild-type (T23), mutant (Mlae1) strains, and a strain overexpressing the *lae1* gene (Olae1). Further, DEGs and metabolites were explored. UPLC-QTOF-MS analysis and bioinformatics analysis were applied to explore the metabolites and specific metabolic pathways associated with the *lae1* gene. The role of *lae1* in the regulation of phospholipid metabolism in *Trichoderma* has not been reported before. Phospholipid metabolism plays an essential role in signal transduction, transmembrane transport, and adaptation to environmental changes. The findings of the current study provide new insights into the role of the *lae1* gene in phospholipid metabolism and further offer a better understanding that regulation of Lae1 on the switch from primary to secondary metabolites improves niche adaptation.

## 2. Materials and Methods

### 2.1. Fungal Strains and Growth Conditions

The wild-type *T. atroviride,* T23 strain (WT, ACCC32730), and *Agrobacterium tumefaciens* strain, AGL 1 were cultured in Culture Collection about Trichoderma of China (CCTC). Two transformants, *lae1* deletion strain (Mlae1) and *lae1* over-expression strain (Olae1), were generated through *Agrobacterium tumefaciens* mediated transformation (ATMT), as described by Yu et al. (2014). The three strains were maintained in potato dextrose agar (PDA) at 28 °C for five days. Three strains (MLae1, T23, OLae1) were inoculated into the middle of PDA plate, and then the sterilized coverslips were inserted into the medium 2–3 cm away from the inoculation site with an angle of 45 angle when the mycelium grew coverslip for 5d, the coverslip was removed and observed under the microscope. For liquid fermentation, a density of 10^6^ spores was inoculated into PD medium and further cultured for five days. The mycelia were then collected and lyophilized using liquid nitrogen and refrigerated for subsequent experiments.

### 2.2. Total RNA Extraction and RNA Sequencing

Total RNA was extracted from freeze-dried mycelium using TRIzol^®^ Reagent (Invitrogen, CV, Waltham, MA, USA) according to the manufacturer’s instructions. Genomic DNA in the samples was removed using DNase I (TaKara, Kusatsu, Japan). RNA quality was determined using a 2100 Bioanalyser (Agilent, Santa Clara, CA, USA) and quantified using ND-2000 NanoDrop (NanoDrop Technologies, Wilmington, DE, USA). High-quality RNA samples were used to construct sequencing library. RNA-seq transcriptome library was prepared with TruSeqTM RNA library preparation Kit from Illumina (San Diego, CA, USA) using 1 μg of total RNA. Messenger RNA was isolated using the polyA selection method by oligo (dT) beads and then fragmented using a fragmentation buffer. Double-stranded cDNA was then synthesized using a SuperScript double-stranded cDNA synthesis kit (Invitrogen, CV) with random hexamer primers (Illumina). The synthesized cDNA was then subjected to end-repair, phosphorylation, and ‘A’ base addition following the Illumina’s library construction protocol. Libraries were size selected for cDNA target fragments of 300 bp on 2% Low Range Ultra Agarose. Selected cDNA target fragments were then amplified by PCR using Phusion DNA polymerase (NEB) for 15 PCR cycles. Paired-end RNA-seq sequencing library was sequenced using Illumina HiSeq xten/NovaSeq 6000 sequencer (2 × 150 bp read length).

### 2.3. Identification of DEGs and Functional Enrichment Analyses

The raw paired-end reads were trimmed, and quality control analysis was performed using SeqPrep (https://github.com/jstjohn/SeqPrep, accessed on 20 February 2019) and Sickle (https://github.com/najoshi/sickle, accessed on 20 February 2019) tools with default parameters. Clean reads were separately aligned to the reference genome using HISAT2 (http://ccb.jhu.edu/software/hisat2/index.shtml, accessed on 20 February 2019) [25] software under orientation mode. Mapped reads from each sample were assembled using StringTie webserver (https://ccb.jhu.edu/software/stringtie/, accessed on 20 February 2019) through a reference-based approach [26]. The expression level of each transcript was determined using the transcripts per million reads (TPM) method to explore DEGs between two different samples. RSEM webserver (http://deweylab.biostat.wisc.edu/rsem/, accessed on 20 February 2019) [27] was used to quantify gene expression levels. Differential expression analysis was performed using DESeq2 [28]/DEGseq [29]/EdgeR [30] tool with a Q value ≤ 0.05. Genes with |log2FC| > 1 and Q value <= 0.05 (DESeq2 or EdgeR)/Q value <= 0.001 (DEGseq) were considered to be significantly differentially expressed genes. Functional-enrichment analysis, including GO and KEGG analyses, were performed to explore the enrichment of DEGs in different GO terms and metabolic pathways. A Bonferroni-corrected *p*-value ≤ 0.05 compared with the whole-transcriptome background was used. GO functional enrichment and KEGG pathway analyses were performed using Goatools (https://github.com/tanghaibao/Goatools, accessed on 20 February 2019) and KOBAS tool (http://kobas.cbi.pku.edu.cn/home.do, accessed on 20 February 2019) [31].

### 2.4. Extraction of Metabolites

Metabolites were extracted from 50 mg mycelia for each sample using a 400 µL methanol: water (4:1, *v/v*) solution. The mixture was allowed to settle at −20 °C then treated with High throughput tissue crusher Wonbio-96c (Shanghai Wanbo Biotechnology Co., Ltd., Shanghai, China) at 50 Hz for 6 min. Further, the mixture was vortexed for 30 s and subjected to ultrasonication at 40 kHz for 30 min at 5 °C. Samples were placed at −20 °C for 30 min to precipitate proteins. Samples were then centrifuged at 13,000× *g* at 4 °C for 15 min, and the supernatant was carefully transferred to sample vials for LC-MS/MS analysis.

### 2.5. UPLC-QTOF-MS/MS Analysis

Chromatographic separation of metabolites was performed on a ExionLC^TM^AD system (AB Sciex, Framingham, MA, USA) equipped with an ACQUITY UPLC BEH C18 column (100 mm × 2.1 mm i.d., 1.7 µm; Waters, Milford, CT, USA). The mobile phase comprised 0.1% formic acid in water (solvent A) and 0.1% formic acid in acetonitrile:isopropanol (1:1, *v/v*) (solvent B). The solvent gradient was programmed as follows: from 0 to 3 min, 95% (A): 5% (B) to 80% (A): 20% (B); from 3 to 9 min, 80% (A): 20% (B) to 5% (A): 95% (B); from 9 to 13 min, 5% (A): 95% (B) to 5% (A): 95% (B); from 13 to 13.1 min, 5% (A): 95% (B) to 95% (A): 5% (B), from 13.1 to 16 min, 95% (A): 5% (B) to 95% (A): 5% (B) to equilibrate the systems. Sample injection volume was set at 20 uL, and the flow rate was set at 0.4 mL/min. The column temperature was maintained at 40 °C. All samples were stored at 4 °C during the period of analysis. A pooled quality control sample (QC) was prepared by mixing equal volumes of all samples as part of the system conditioning and quality control process.

The UPLC system was coupled with a quadrupole-time-of-flight mass spectrometer (Triple TOF^TM^5600+, AB Sciex, USA) equipped with an electrospray ionization (ESI) source. The detector was operated under a positive mode and negative mode. Optimal conditions for the detector were set as follows: source temperature, 500 °C; curtain gas (CUR), 30 psi; both Ion Source GS1 and GS2, 50 psi; ion-spray voltage floating (ISVF), −4000 V under negative mode and 5000 V under positive mode, respectively; declustering potential, 80 V; collision energy (CE), 20–60 V rolling for MS/MS. Data acquisition was performed with the Data Dependent Acquisition (DDA) mode. Detection of metabolites was performed over a mass range of 50–1000 m/z. Mean values from six biological replicates were obtained. Statistical analysis was performed on log-transformed data to identify significant differences in metabolite levels between groups. The identification of metabolites was performed using the accurate mass, MS/MS fragments spectra, and isotope ratio difference and comparing the data to relevant biochemical databases, such as Human metabolome database (HMDB) (http://www.hmdb.ca/, accessed on 21 December 2018) and Metlin database (https://metlin.scripps.edu/, accessed on 21 December 2018).

### 2.6. Multivariate Statistical Analysis

Multivariate analysis was performed using ropls (Version1.6.2, http://bioconductor.org/packages/release/bioc/html/ropls.html, accessed on 21 December 2018) R package from Bioconductor using the Majorbio Cloud Platform (https://cloud.majorbio.com, accessed on 21 December 2018). Principle component analysis (PCA) was performed on metabolite data using an unsupervised method. General clustering, trends, or outliers of metabolite profiles were visualized using PCA plots. All metabolite variables were scaled to unit variances before performing PCA. Orthogonal partial least squares discriminate analysis (OPLS-DA) determines global metabolic changes between comparable groups. Metabolite variables were scaled to Pareto Scaling before conducting the OPLS-DA. The validity of the model was evaluated using model parameters R2 and Q2, which provide information for interpretability and predictability of the model, respectively, to minimize the risk of over-fitting. Variable importance in the projection (VIP) was determined using OPLS-DA model. P-values were estimated with paired Student’s *t*-test for single-dimensional analysis.

### 2.7. Differential Metabolite Analysis

VIP value more than 1 and *p*-value less than 0.05 denoted statistical significance among groups. A total of 92 differential peaks were selected. Differential metabolites among the two groups were summarized and mapped into their biochemical pathways through metabolic enrichment and pathway analyses based on database search (KEGG, http://www.genome.jp/kegg/, accessed on 21 December 2018). The metabolites were then classified according to the pathways they were involved in or the functions they performed. Enrichment analysis was performed to explore a group of metabolites in a function node. The principle was that annotation analysis of a single metabolite develops into annotation analysis of a group of metabolites. Statistically significantly enriched pathways were explored using scipy.stats (Python package) (https://docs.scipy.org/doc/scipy, accessed on 21 December 2018/) by performing Fisher’s exact test.

## 3. Results

### 3.1. Effects of Lae1 on Phenotype in Trichoderma atroviride 23

To investigate the impact of LAE1 on the development of *T. atroviride, lae1* null mutants (Mlae1) were generated by replacing the *lae1* coding region with the hygromycin B phosphotransferase gene *hph*. In addition, we generated overexpressing (Olae1) mutants by fusing the *lae1* ORF downstream of the strongly expressing TrpC promoter. The transformants were purified and verified by PCR (Appendix A). As shown in Figure 1, after *the lae1* gene was knocked out, the mycelium and pigment were significantly reduced, and T23 and T23*Olae1* grew more densely and secreted an amount of pigment than that of Mlae1(Figure 1A) on the PDA plate. In addition, the over-expression of *lae1* exhibited a higher conidiation density compared with that of the Mlae1 and T23 strains (Figure 1B). This suggests that *lae1* impacts the hyphal morphology and spore production of *T. atroviride.*

### 3.2. Profiling the Transcriptome of Three Strains: Analysis of Differentially Expressed Genes (DEGs)

To understand genes regulated by *lae1* at the transcriptional level. Strain Mlae1, T23, and Olae1 were collected and subjected to RNA-seq and compared to the difference in gene expression. The results showed that a total of 7310, 8878, and 7883 genes had relative expression levels above one transcript per million (TPM) in Mlae1, T23, and Olae1, respectively. Notably, 6975 genes were expressed in the three strains, whereas 66, 1107, and 171 genes were uniquely expressed in Mlae1, T23, and Olae1, respectively (Appendix A). Genes with *p*-values < 0.05 and FC (fold change) > 2 were selected to represent significantly differentially expressed genes of the mutant or overexpressed strains relative to the wild-type. The results showed that 2985 genes were significantly downregulated, whereas 2361 genes were significantly upregulated in Mlae1. Further, 2135 genes were significantly downregulated, whereas 2276 genes were significantly upregulated in Olae1 (Appendix A). Interestingly, most of the significantly downregulated/upregulated genes in the mutant were not significantly upregulated/downregulated in the overexpressed strains, and vice versa. For example, only 18 genes were significantly upregulated in Mlae1 and significantly downregulated in Olae1. On the contrary, 1498 genes were significantly upregulated in Mlae1 and Olae1, whereas 1742 genes were significantly downregulated in Mlae1 and Olae1. Deletion and over-expression of the *lae1* gene resulted in similar effects, indicating that other molecules possibly interact with *lae1* and play a regulatory role together.

### 3.3. GO Functional Analysis of Differentially Expressed Genes

GO annotation analysis was performed to explore the classes of genes regulated by *lae1*. The results showed that DEGs were functionally similar in the mutant and overexpressed strains. Molecular function ontology results showed that the most enriched terms were catalytic activity and binding in the molecular function category. Membrane part, cell part, and organelle were the most enriched terms in the cellular component ontology. Metabolic and cellular processes were the most enriched terms in the biological process category (Figure 2). Metabolic processes accounted for a large proportion of the four groups of DEGs. Overall, results indicated that most DEGs regulate metabolic processes through changing catalytic activity or binding, and these DEGs mainly function in the proximity of the membrane and within organelles.

GO enrichment analysis was performed based on DEGs between Mlae1 and T23 (Figure 3). GO terms significantly enriched in the Mlae1 included the ncRNA metabolic process, rRNA metabolic process, organic cyclic compound biosynthetic process, and aromatic compound biosynthetic process. Downregulation of these genes may indicate a deficiency of Mlae1 in the previously mentioned terms. Further, KEGG enrichment analysis was performed to explore metabolic pathways regulated by the DEGs. DEGs between Mlae1 and T23 were mainly implicated in regulating ribosome biogenesis in eukaryotes, glycolysis/gluconeogenesis, the TCA cycle, and sphingolipid metabolism (Figure 4). Moreover, DEGs identified between Olae1 and T23 were involved in the TCA cycle, propanoate metabolism, oxidative phosphorylation, glycolysis/gluconeogenesis, and glycerophospholipid metabolism.

### 3.4. Metabolome Difference Analysis

Principal component analysis showed significant differences among the three strains (Appendix A). A discriminant analysis approach (orthogonal projections to latent structures-discriminant analysis, OPLS-DA) was applied to obtain classification models and explore significantly differential metabolites (Appendix A). Metabolites with VIP > 1 and *p*-value < 0.05 were selected for subsequent analysis. A total of 92 metabolites presented significant differences after comparing T23 with Mlae1 and Olae1 strains. The results showed significantly higher levels of 45 and 17 compounds in Mlae1 and Olae1, respectively, compared with the levels in the T23 strain. The chemical compounds with significant variations included amino acids, organic acids, alcohols, aldehydes, phospholipids, and alkaloids.

### 3.5. Analysis of Differential Metabolites

To visualize the differences among the three strains, we selected the top 50 metabolites according to their abundances. Subsequently, hierarchical clustering analysis (HCA) was performed, and a heat map was generated to visualize the differences among the three strains. The heat map showed a clear clustering, demonstrating the reliability of the OPLS-DA models in distinguishing the different metabolomes among Mlae1, Olae1, and T23 (Figure 5).

VIP score was used to rank the contribution of metabolites to the model according to the weighting coefficient of the OPLS-DA model (Figure 6). The metabolites with the highest VIP value between T23 and Mlae1 were phospholipids: PI(13:0/18:4(6Z,9Z,12Z,15Z)) and PS PS(DiMe(9,5)/MonoMe(11,3)). Expression of PI(13:0/18:4(6Z,9Z,12Z,15Z)) was upregulated in Mlae1, whereas PS(DiMe(9,5)/MonoMe(11,3)) expression was upregulated in T23 strain. The expression of phospholipids was significantly different between wild and mutant strains. The contents of PI and LysoPC were higher in the mutant strain, whereas the contents of PS and LysoPE were higher in the wild-type strain. The polar head space of PI and PC is larger relative to that of PS and PE, which may result in differences in curvature and area of the cell membrane between Mlae1 and T23 [32].

Further analysis of the compounds with significant variations showed that some compounds had unique functions (Table 1 and Table 2). For instance, 15-Acetyl-4-deoxynivalenol, ACRL Toxin II, Gliricidol, and Acetyldeoxynivalenol are mycotoxins. Mycotoxins are secondary metabolites from fungi that are harmful to other organisms. The levels of these mycotoxins were significantly higher in the mutant strain, Mlae1, compared with the wild-type strain. Levels of drug compounds were significantly different in Mlae1 compared with the level in the T23 strain. Nelfinavir is a protease inhibitor used to abrogate viral replication and improve immune function in HIV-infected individuals [33]. The yield of Nelfinavir was lower in Mlae1 relative to that in T23. Moreover, Nabumetone, a nonsteroidal anti-inflammatory prodrug [34], and Rotigotine, used for the treatment of idiopathic Parkinson’s disease, were identified [35]. Sapidolide A exhibits strong inhibitory activity against pathogenic fungi [36]. Sapidolide A showed a significantly higher yield in Mlae1 compared with the T23 strain. Notably, Josamycin and Maduramicin antibiotics were detected in the Olae1 strain.

### 3.6. KEGG Analysis of Differential Metabolites

KEGG pathway analysis was used to explore the enrichment of the 92 significantly expressed metabolites in different pathways. Relative-betweenness centrality was used as the topological method, then Bonferroni correction was performed (Figure 7 and Appendix A). The results showed that three metabolite pathways had the most significant variations based on the corrected *p*-value (*p*-value < 0.05). The three pathways were central carbon metabolism in cancer, choline metabolism in cancer, and glycerophospholipid metabolism. Metabolites related to these pathways were L-Tyrosine, PE (14:1(9Z)/22:2(13Z,16Z)), Glycerophosphocholine, Isocitrate, L-Malic acid, LysoPC (18:2(9Z,12Z)), and LysoPC (20:5(5Z,8Z,11Z,14Z,17Z)). A previous study indicated that Lae1 was known to function as an activator of the secondary metabolism of Trichoderma. Our study indicated that *lae1* also regulates lipid metabolism, central carbon metabolism, and amino acid metabolism. These data showed that the *Lae1* gene potentially regulates the switch between primary and secondary metabolism.

## 4. Discussion

### 4.1. Integrated Analysis of Metabolic Pathways

Significantly altered metabolic profiles between Mlae1 and T23 strains were explored in the present study. The results indicated that the TCA cycle, phospholipid metabolism, and secondary metabolism of Mlae1 were the main metabolic-affected pathways. Gene expression results showed that the cycle from isocitrate to malate was upregulated; however, the decrease in isocitrate level and downregulation of the *ACO* gene indicate that the TCA cycle was repressed at the initial step. Changes in secondary metabolism and repression of tyrosine metabolism may be associated with the repression of the TCA cycle.

It is the first report that Mlae1 significantly affected the gene expression of Lipid metabolism. The pathway for conversion of acetyl-CoA and sn-glycerol-3P to diacyl-sn-glycerol-3P was significantly upregulated in Male1. Upregulation of *CDIPT* and *CHO1* genes promoted the synthesis of PI and PS, respectively. Moreover, the upregulation of *psd* and *plc* promoted the conversion of PS to PE and further to Diacyl-sn-glycerol, which is the synthetic precursor of PC. Downregulation of the *PLB* gene, which is implicated in regulating the formation of glycerophosphocholine from LysoPC, caused a significant increase in LysoPC and a decrease in glycerophosphocholine levels in Mlae1.

### 4.2. Lae1 Regulates Phospholipid Metabolism

Phospholipids are water-insoluble molecules with diverse functions within cells. They are involved in maintaining electrochemical gradients, subcellular partitioning, first- and second-messenger cell signaling protein trafficking, and membrane anchoring [37]. Glycerolipids, including phosphatidylcholine (PC), phosphatidylserine (PS), phosphatidylethanolamine (PE), and phosphatidylinositol (PI), are the basic component of biofilms. The major classes of phospholipids identified in *T. harzianum* cultures were PC and PE [38]. The changes in the content of PE and PC influenced the PC/PE ratio. An increase in the PC/PE ratio stabilizes the lipid bilayer, while its decrease may increase membrane permeability, which in turn may disturb the integrity of the cell. Notably, some glycerolipids, such as PI, play essential roles in cell signal transduction.

Among the 92 differentially expressed metabolites, 21 different phospholipids were identified, accounting for 22.8% of the total differential metabolites. This indicates that *lae1* is implicated in the regulation of phospholipid metabolism. This finding was verified at the gene level through transcriptome analysis. The analysis showed that DEGs between Mlae1 and T23 were associated with the enrichment of sphingolipid metabolism and glycerolipid metabolism. In addition, KEGG enrichment analysis of DEGs between Olae1 and T23 showed enrichment of glycerophospholipid metabolism, implying that *lae1* regulates phospholipid metabolism. Deletion of the *lae1* gene changes the contents and proportions of various glycerolipids in cells. These changes potentially impact the intracellular membrane area and membrane curvature, ultimately affecting metabolic pathways.

### 4.3. Lae1 Gene Regulates Phosphatidylinositol Signaling System

The effects of significant changes in phospholipid profiles were not clear. One of the possible explanations is that increase in PI in Mlae1 promoted the phosphatidylinositol signaling system. Analysis showed the presence of significantly expressed genes between Mlae1 and T23 implicated in the regulation of the phosphatidylinositol signaling system (Appendix A). PI(4,5)P2 is an important phosphate ester derivative in biological cells. The findings showed that phosphatidylinositol 4-kinase (*PI4KB*) and phosphatidylinositol 5-kinase (*PIKFIVE*) were downregulated, indicating a decrease in the content of PI(4,5)P2 in Mlae1. Phospholipase-C (PLC) acts on PI(4,5)P2 to form water-soluble inositol triphosphate (IP3) and fat-soluble diacylglycerol(DG), which are two important second messengers, thus initiating and activating intracellular calcium signaling pathways [39] and phospholipase-C gene D (plcD) was downregulated in Mlae1. CaM is the main Ca^2+^ receptor in cells [40], and the results of the current study showed that the *Calm* gene was upregulated in the Mlae1 strain. Genes involved in the regulation of signal initiation were downregulated in the Mlae1 strain, whereas genes downstream of the regulatory signaling system were upregulated. These findings indicate a possible feedback regulation mechanism.

### 4.4. Changes in Phospholipid Metabolism Affects Membrane Composition

Phospholipids are important components of membrane bilayers. PC, PE, PI, and PS are major components of membranes. These four phospholipids are asymmetrically distributed across the membrane. PC is mainly distributed in the outer leaflet, whereas PE is localized in the inner leaflet [41]. The findings of the present study indicated that PC/PE ratio was significantly high in Mlae1 compared with T23. This increase in PC/PE ratio may improve the stability of the membrane in the Mlae1 strain because PE forms a hexagonal phase to reduce membrane fluidity [42]. A study showed a decrease in the PC/PE ratio, possibly leading to a decrease in membrane fluidity in the biomass of *Trichoderma harzianum* IM 0961 exposed to 2,4-D [38]. This suggests that microorganisms use different strategies to alter phospholipid membranes to maintain cell stability in the presence of various stress environmental conditions.

Several membrane proteins are distributed on the membrane in addition to phospholipids. Studies have shown that most membrane proteins interact with phospholipids [43,44,45]. GO annotation analysis showed that DEGs were associated with the enrichment of the membrane term. Moreover, KEGG enrichment analysis of DEGs indicated that oxidative phosphorylation was highly enriched. These findings indicate that the changes in composition and level of phospholipids are related to the changes in membrane proteins. Changes in the proportion of membrane lipids may affect the localization of membrane proteins, or changes in the function of membrane proteins may require specific composition of phospholipids. Changes in composition and levels of membrane lipids and membrane proteins potentially alter membrane functions, such as proton transfer, oxidative phosphorylation, and transport of other metabolites.

### 4.5. ATP Synthase Genes Were Upregulated in Mlae1

ATP synthesis is accomplished by a variety of mitochondrial membrane proteins. Transcriptome analysis showed several differentially expressed genes involved in ATP synthase in the Mlae1 strain (Appendix A). Notably, most of these genes were significantly upregulated in Mlae1 compared with the expression levels in the T23 strain. *Atp6V0A1* and *atp6c* genes code for A- and V-type subunits of ATPase, respectively. Moreover, several genes related to the mitochondrial ATPase subunit were differentially expressed. Upregulation of these genes significantly increased the rate of ATP synthesis in the Mlae1 strain. Furthermore, some genes involved in ribosome synthesis were significantly upregulated in Mlae1. These findings imply that high ATP levels were used to increase RNA processing and protein translation in the Mlae1 strain. In addition, an increase in expression levels of ATP synthase on the membrane may be related to the changes in phospholipid profiles.

Since *lae1* is conserved global regulation on the synthesis of secondary metabolites, its role in the regulation of phospholipid metabolism may be of considerable relevance to the synthesis and secretion of secondary metabolites. Our findings may provide new insights into the physiological functions of Lae1 and its regulation of lipid metabolism.

## 5. Conclusions

The evolution of diverse metabolic strategies in *Trichoderma* has enabled it to adapt to fluctuating environmental changes, conferring high fitness and the ability to withstand stress at multiple levels [46].

In the current study, we explored the function of the *lae1* gene through metabolomic and transcriptomic analysis. The results indicate that *the lae1* gene affects Acetyl-CoA centered carbon metabolism and amino acid metabolism. Moreover, the *lae1* gene modulates the synthesis of secondary metabolites. The findings of this study revealed significantly altered profiles of several secondary metabolites with antibacterial activity. In addition, the metabolism and synthesis of phospholipids were affected by changes in expression levels of *lae1*, which has not been reported previously. Changes in phospholipid metabolism may affect the composition of cell membranes, phosphatidylinositol signaling system, and oxidative phosphorylation. The *Lae1* gene potentially regulates the switch between primary and secondary metabolism. The discovery of *lae1* as a phospholipid regulator further extends the role of *lae1* in the qualitative and quantitative composition of phospholipids composites to improve the membrane permeability to increase the secretion of metabolites and against the stress habitats.

Overall, our results extend our understanding of the *lae1* regulation in lipid metabolism; these findings could potentially provoke in-depth investigations of the Lae1-mediated target genes in phospholipid synthesis.

Data were analyzed using Majorbio Cloud Platform (www.majorbio.com, accessed on 20 February 2019).

## Figures and Tables

**Figure 1 jof-09-00120-f001:**
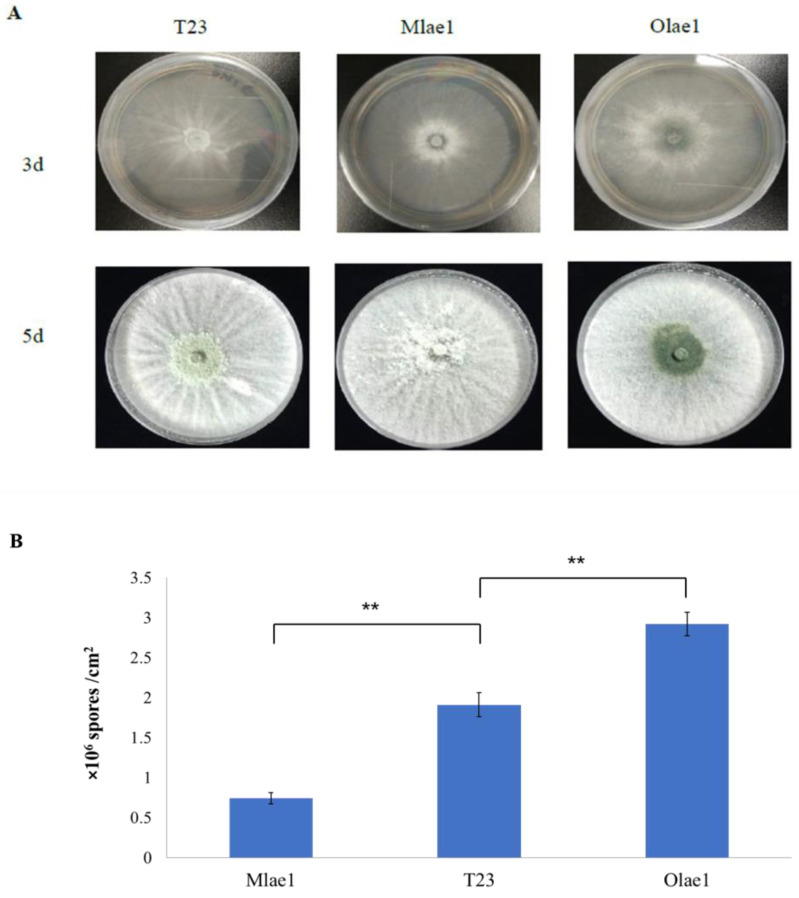
Growth phenotype of the three strains cultured on (**A**) PDA medium for 3d and 5d and (**B**) number of spores in the three strains (T23, Mlae1 and Olae1). Significance level determined using *t*-test; denoted by **: *p* < 0.01.

**Figure 2 jof-09-00120-f002:**
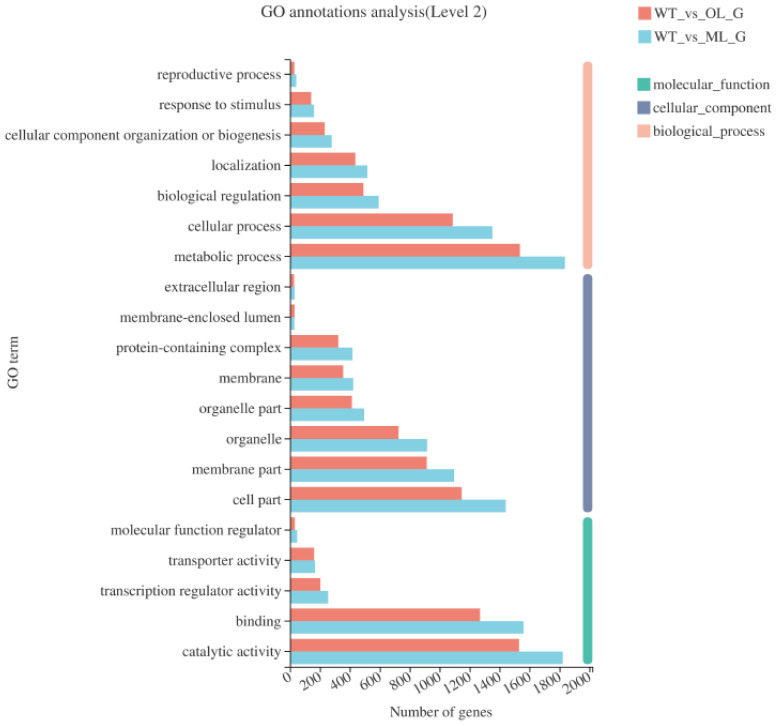
GO function analysis of DEGs. WT_vs_OL_G represents DEGs between Olae1 and T23 wild-type strain, WT_vs_ML_G represents DEGs between Mlae1 and T23 wild-type strain.

**Figure 3 jof-09-00120-f003:**
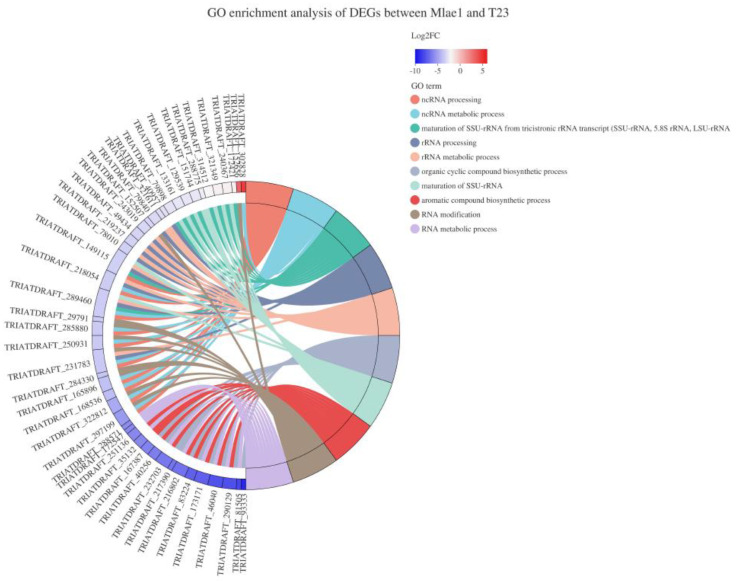
GO enrichment analysis of DEGs between Mlae1 and T23 strains.

**Figure 4 jof-09-00120-f004:**
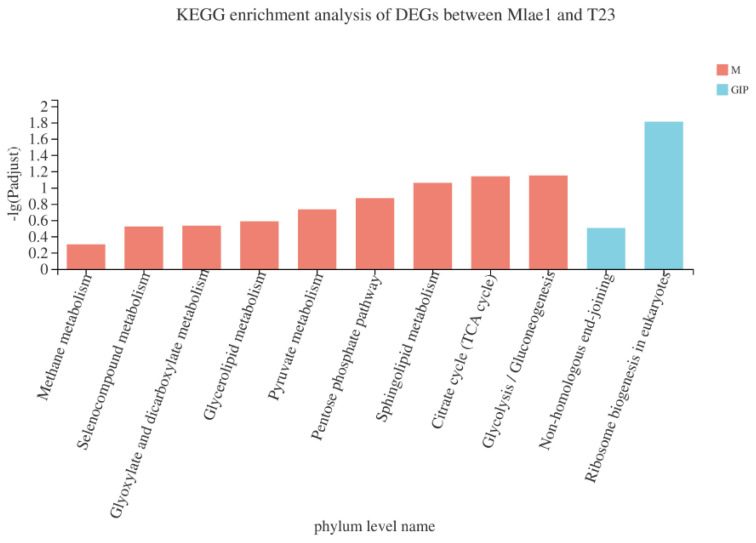
KEGG enrichment analysis of DEGs between Mlae1 and T23 strains. M refers to the Metabolism pathways in KEGG. GIP refers to the Genetic Information Processing pathways in KEGG.

**Figure 5 jof-09-00120-f005:**
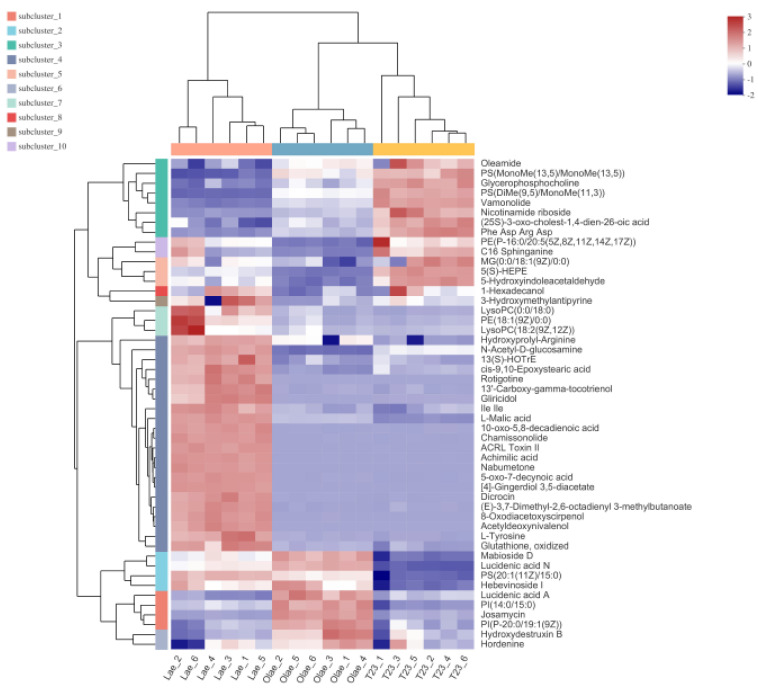
Hierarchical clustering analysis for identification of differently expressed metabolites. Each column in the figure represents a sample, each row represents a metabolite, and the color indicates relative amount of metabolites expressed in the group.

**Figure 6 jof-09-00120-f006:**
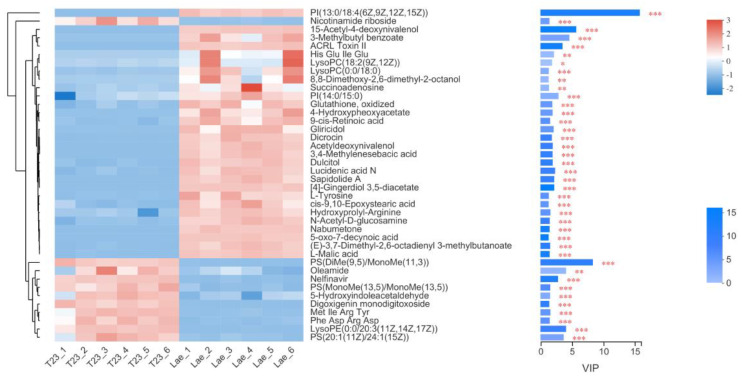
Variable importance in projection (VIP) scores of significantly different metabolites between T23 and Mlae1 strains.

**Figure 7 jof-09-00120-f007:**
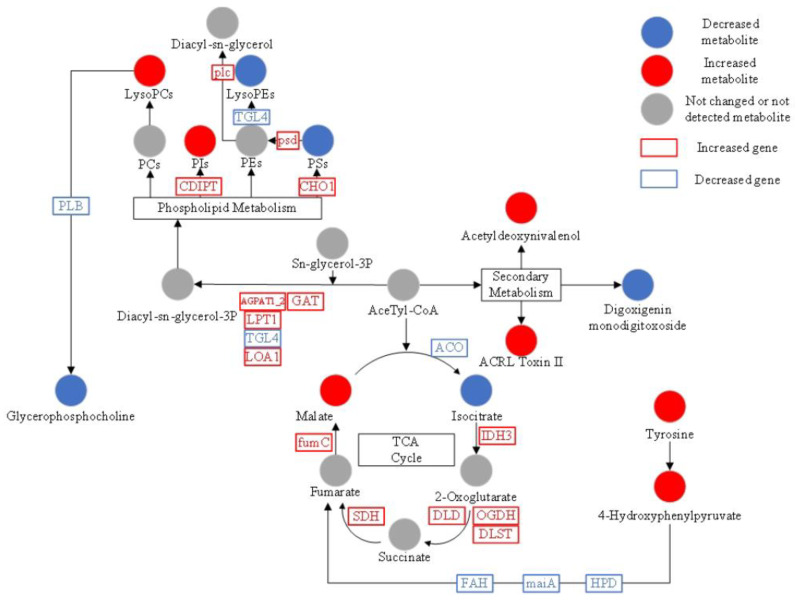
Overview of significantly altered metabolic profiles in Mlae1 and T23 strains. Red and blue colors represent upregulated and downregulated metabolites in Mlae1 relative to the level in T23, respectively.

**Table 1 jof-09-00120-t001:** Key metabolites with significant differences between T23 and Mlae1 strains.

Metabolite	Formula	VIP	FC (T23/Mlae1)	*p*-Value
15-Acetyl-4-deoxynivalenol	C_17_H_22_O_7_	5.6182	0.0059	1.83 × 10^−15^
ACRL Toxin II	C_17_H_24_O_5_	3.4046	0.0013	1.43 × 10^−14^
Nelfinavir	C_32_H_45_N_3_O_4_S	2.6966	2.6108	5.47 × 10^−13^
Sapidolide A	C_14_H_18_O_5_	2.089	0.0001	1.91 × 10^−10^
Gliricidol	C_16_H_16_O_7_	1.9971	0.0135	1.21 × 10^−6^
Acetyldeoxynivalenol	C_17_H_22_O_7_	1.8362	0	2.28 × 10^−10^
Nabumetone	C_15_H_16_O_2_	1.347	0.0005	4.98 × 10^−14^
Digoxigenin monodigitoxoside	C_29_H_44_O_8_	1.2926	4.6022	3.73 × 10^−11^
cis-9,10-Epoxystearic acid	C_18_H_34_O	1.1944	0.4748	5.80 × 10^−7^
Rotigotine	C_19_H_25_NOS	1.0502	0.0006	3.85 × 10^−8^

**Table 2 jof-09-00120-t002:** Key metabolites with significant differences between T23 and Olae1 strains.

Metabolite	Formula	VIP	FC (T23/Olae1)	*p*-Value
Josamycin	C_42_H_69_NO_15_	2.5971	0.244	9.84 × 10^−11^
Maduramicin	C_47_H_80_O_17_	1.0745	0.2895	1.25 × 10^−10^
Lucidenic acid A	C_27_H_38_O_6_	2.0008	0.5255	1.00 × 10^−6^
Hydroxydestruxin B	C_30_H_51_N_5_O_8_	1.1761	0.6212	1.66 × 10^−3^
Ergothioneine	C_9_H_15_N_3_O_2_S	1.1284	1.387	2.15 × 10^−5^
Isocitrate	C_6_H_8_O_7_	2.1805	1.4663	8.75 × 10^−11^
Pentamidine	C_19_H_24_N_4_O_2_	1.1724	1.8769	4.26 × 10^−8^

## Data Availability

Data is contained within the article or Appendix A. The data presented in this study are available in Majorbio Cloud Platform (https://cloud.majorbio.com/page/tools/).

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
