# Peer review of "Integrated Transcriptome and Untargeted Metabolomic Analyses Revealed the Role of Methyltransferase Lae1 in the Regulation of Phospholipid Metabolism in Trichoderma atroviride"

_jof, 2023, doi:10.3390/jof9010120_

Round 1

Reviewer 1 Report

In this paper, the authors performed an integrated analysis of transcriptomics and untargeted metabolomics in Trichoderma atroviride, and revealed that methyltransferase Lae1 modulated the regulation of phospholipid metabolism. There are some questions that I think help improve this manuscript.

1. In Figure 1B, statistical analysis is necessary.

2. Analysis of comparative transcriptomes requires to address in detail.

3. The reasonable experiments for further confirming transcriptomic and metabolomics are necessary.

4. In discussion section, the association analysis among strain phenotypes, gene expression and metabolites should be addressed in detail.

5. Some language expressions need to be improved.

Author Response

Dear reviewer,

   Thanks for your good question. I have give response to your advice. Please see the attachment. 

Sincerely,

Yaqian

Reviewer 2 Report

It is an interesting work which evaluated Lae 2 function by comparing the transcriptome and metabolome of Trichoderma atroviride WT (T23),deletion mutant (Mlae1) and overexpressing lae1 strain (Olae1). This has been done in previous studies for Trichoderma and perhaps the novelty is the one used by RNA-seq and metabolomics for a more global understanding of the function of Lae1.

In other to clarify and input the MS I have some comments and suggestions:

General Questions:

-          Which is the novelty of the work as compared with previous studies that also used Trichoderma mutants for dilucidated Lae1 role?

-          Is lae1 1 expression influenced by the carbon source?. Why did you use a complex media like PDB for growth?

Introduction

Since you consider your work is fist reporting the function of Lae 1 in phospholipid metabolism, could you include some supporting references for stress the novelty?

About M&M

-          Figure 1B shows the spore density (spore/cm2) in the strains evaluated. Could you explain and put in the corresponding section how do you perform this analysis?

-          Has the transcriptome been deposited in any data base? Dou you have the bioproject number?

-          Tables S1 and S2 did not mentioned in the MS. These data correspond to qPCR validation the transcriptomic data? Please include the title, and the corresponding gene function in an additional column.

Results

-          In the figure 1A, you shows differences on colony pigmentation between the three evaluated strain. The lack of pigments in Mlae 1 means non sporulation? Is could be possible that the colony presented albino spores? Did you check this by microscopy?

-          About Figure 1B, please check Y axis values. For example, for the Mlae1 strain, you report less than 1 spores per /cm2. This makes sense?

-          Figure 1 appeared with capital letters (i.e Figure 1A) but is referred with lowercase letters in the text (i.e Figure 1a).

-          In the figure 4, please indicates the corresponding legend. What GIP means?

-          Supplementary figures S1-S4 are presented neither title nor legend. It difficult an adequate interpretation of the corresponding data.

-          For figures 3,4,6. Why did you exclude the data for Olae1 strain?

-          Figure 7 show that some metabolites belonging to phospholipid metabolism accumulated or decreased in Mlae1 strain. What happened with the corresponding analysis for Olae1 strain (compared with the wild strain) Did you find any correlation?

Discussion

-          According to the involved metabolic pathways, how the loss of lae1 (Mlae1) could induces mycotoxin synthesis? It could be related with the loss of pigmentation.

-          Since there is not information about the metabolome of Olae1 strain related to phospholipid metabolism, I am not convinced about the role of Lae1 in this aspect.

General observations

Line 206 : check transcriptom

Line 402: check synthesiss

Author Response

Dear reviewer,

    Thanks for you good suggestions. We  think carefully and revised the manuscript on the basis of your suggestion.  We hope that  we further carry out some related experiment in furture and get more  novel feature for Lae1 regulator.

Sincerely

Yaqian

Round 2

Reviewer 1 Report

No any comment